# An Off-policy Policy Gradient Theorem Using Emphatic Weightings

**Ehsan Imani,**[*] **Eric Graves,**[*] **Martha White**
Reinforcement Learning and Artificial Intelligence Laboratory
Department of Computing Science
University of Alberta
{imani,graves,whitem}@ualberta.ca

## Abstract

Policy gradient methods are widely used for control in reinforcement learning, particularly for the continuous action setting. There have been a host of theoretically sound algorithms proposed for the on-policy setting, due to the existence of the policy gradient theorem which provides a simplified form for the gradient. In off-policy learning, however, where the behaviour policy is not necessarily attempting to learn and follow the optimal policy for the given task, the existence of such a theorem has been elusive. In this work, we solve this open problem by providing the first off-policy policy gradient theorem. The key to the derivation is the use of *emphatic weightings*. We develop a new actor-critic algorithm—called Actor Critic with Emphatic weightings (ACE)—that approximates the simplified gradients provided by the theorem. We demonstrate in a simple counterexample that previous off-policy policy gradient methods—particularly OffPAC and DPG—converge to the wrong solution whereas ACE finds the optimal solution.

## 1 Introduction

Off-policy learning holds great promise for learning in an online setting, where an agent generates a single stream of interaction with its environment. *On-policy* methods are limited to learning about the agent's current policy. Conversely, in *off-policy* learning, the agent can learn about many policies that are different from the policy being executed. Methods capable of off-policy learning have several important advantages over on-policy methods. Most importantly, off-policy methods allow an agent to learn about many different policies at once, forming the basis for a predictive understanding of an agent's environment [Sutton et al., 2011, White, 2015] and enabling the learning of options [Sutton et al., 1999, Precup, 2000]. With options, an agent can determine optimal (short) behaviours, starting from its current state. Off-policy methods can also learn from data generated by older versions of a policy, known as experience replay, a critical factor in the recent success of deep reinforcement learning [Lin, 1992, Mnih et al., 2015, Schaul et al., 2015]. They also enable learning from other forms of suboptimal data, including data generated by human demonstration, non-learning controllers, and even random behaviour. Off-policy methods also enable learning about the optimal policy while executing an exploratory policy [Watkins and Dayan, 1992], thereby addressing the exploration-exploitation tradeoff.

Policy gradient methods are a general class of algorithms for learning optimal policies, for both the on and off-policy setting. In policy gradient methods, a parameterized policy is improved using gradient ascent [Williams, 1992], with seminal work in actor-critic algorithms [Witten, 1977, Barto et al., 1983] and many techniques since proposed to reduce variance of the estimates of this gradient [Konda and Tsitsiklis, 2000, Weaver and Tao, 2001, Greensmith et al., 2004, Peters et al., 2005,

---

[*]These authors contributed equally.

Bhatnagar et al., 2008, 2009, Grondman et al., 2012, Gu et al., 2016]. These algorithms rely on a fundamental theoretical result: the *policy gradient theorem*. This theorem [Sutton et al., 2000, Marbach and Tsitsiklis, 2001] simplifies estimation of the gradient, which would otherwise require difficult-to-estimate gradients with respect to the stationary distribution of the policy and potentially of the action-values if they are used.

Off-policy policy gradient methods have also been developed, particularly in recent years where the need for data efficiency and decorrelated samples in deep reinforcement learning require the use of experience replay and so off-policy learning. This work began with OffPAC [Degris et al., 2012a], where an off-policy policy gradient theorem was provided that parallels the on-policy policy gradient theorem, but only for tabular policy representations.[2] This motivated further development, including a recent actor-critic algorithm proven to converge when the critic uses linear function approximation [Maei, 2018], as well as several methods using the approximate off-policy gradient such as Deterministic Policy Gradient (DPG) [Silver et al., 2014, Lillicrap et al., 2015], ACER [Wang et al., 2016], and Interpolated Policy Gradient (IPG) [Gu et al., 2017]. However, it remains an open question whether the foundational theorem that underlies these algorithms can be generalized beyond tabular representations.

In this work, we provide an off-policy policy gradient theorem, for general policy parametrization. The key insight is that the gradient can be simplified if the gradient in each state is weighted with an emphatic weighting. We use previous methods for incrementally estimating these emphatic weightings [Yu, 2015, Sutton et al., 2016] to design a new off-policy actor-critic algorithm, called Actor-Critic with Emphatic weightings (ACE). We show in a simple three state counterexample, with two states aliased, that solutions are suboptimal with the (semi)-gradients used in previous off-policy algorithms—such as OffPAC and DPG. We demonstrate both the theorem and the counterexample under stochastic and deterministic policies, and that ACE converges to the optimal solution.

## 2    Problem Formulation

We consider a Markov decision process $(\mathcal{S}, \mathcal{A}, P, r)$, where $\mathcal{S}$ denotes the finite set of states, $\mathcal{A}$ denotes the finite set of actions, $P : \mathcal{S} \times \mathcal{A} \times \mathcal{S} \to [0, \infty)$ denotes the one-step state transition dynamics, and $r : \mathcal{S} \times \mathcal{A} \times \mathcal{S} \to \mathbb{R}$ denotes the transition-based reward function. At each timestep $t = 1, 2, \ldots$, the agent selects an action $A_t$ according to its behaviour policy $\mu$, where $\mu : \mathcal{S} \times \mathcal{A} \to [0, 1]$. The environment responds by transitioning into a new state $S_{t+1}$ according to $P$, and emits a scalar reward $R_{t+1}$ such that $\mathbb{E}[R_{t+1}|S_t, A_t, S_{t+1}] = r(S_t, A_t, S_{t+1})$.

The discounted sum of future rewards given actions are selected according to some target policy $\pi$ is called the return, and defined as:

$$G_t \overset{\text{def}}{=} R_{t+1} + \gamma_{t+1}R_{t+2} + \gamma_{t+1}\gamma_{t+2}R_{t+3} + \ldots \tag{1}$$
$$= R_{t+1} + \gamma_{t+1}G_{t+1}$$

We use transition-based discounting $\gamma : \mathcal{S} \times \mathcal{A} \times \mathcal{S} \to [0, 1]$, as it unifies continuing and episodic tasks [White, 2017]. Then the state value function for policy $\pi$ and $\gamma$ is defined as:

$$v_\pi(s) \overset{\text{def}}{=} \mathbb{E}_\pi[G_t|S_t = s] \quad \forall s \in \mathcal{S} \tag{2}$$
$$= \sum_{a \in \mathcal{A}} \pi(s, a) \sum_{s' \in \mathcal{S}} P(s, a, s')[r(s, a, s') + \gamma(s, a, s')v_\pi(s')] \quad \forall a \in \mathcal{A}, \forall s \in \mathcal{S}$$

In off-policy control, the agent's goal is to learn a target policy $\pi$ while following the behaviour policy $\mu$. The target policy $\pi_{\boldsymbol{\theta}}$ is a differentiable function of a weight vector $\boldsymbol{\theta} \in \mathbb{R}^d$. The goal is to learn $\boldsymbol{\theta}$ to maximize the following objective function:

$$J_\mu(\boldsymbol{\theta}) \overset{\text{def}}{=} \sum_{s \in \mathcal{S}} d_\mu(s)i(s)v_{\pi_{\boldsymbol{\theta}}}(s) \tag{3}$$

where $i : \mathcal{S} \to [0, \infty)$ is an interest function, $d_\mu(s) \overset{\text{def}}{=} \lim_{t \to \infty} P(S_t = s|s_0, \mu)$ is the limiting distribution of states under $\mu$ (which we assume exists), and $P(S_t = s|s_0, \mu)$ is the probability that

$S_t = s$ when starting in state $s_0$ and executing $\mu$. The interest function—introduced by Sutton et al. [2016]—provides more flexibility in weighting states in the objective. If $i(s) = 1$ for all states, the objective reduces to the standard off-policy objective. Otherwise, it naturally encompasses other settings, such as the start state formulation by setting $i(s) = 0$ for all states but the start state(s). Because it adds no complications to the derivations, we opt for this more generalized objective.

## 3 Off-Policy Policy Gradient Theorem using Emphatic Weightings

The policy gradient theorem with function approximation has only been derived for the on-policy setting thus far, for stochastic policies [Sutton et al., 2000, Theorem 1] and deterministic policies [Silver et al., 2014]. The policy gradient theorem for the off-policy case has only been established for the setting where the policy is tabular [Degris et al., 2012b, Theorem 2].[3] In this section, we show that the policy gradient theorem does hold in the off-policy setting, when using function approximation for the policy, as long as we use emphatic weightings. These results parallel those in off-policy policy evaluation, for learning the value function, where issues of convergence for temporal difference methods are ameliorated by using an emphatic weighting [Sutton et al., 2016].

**Theorem 1** (Off-policy Policy Gradient Theorem).

$$\frac{\partial J_\mu(\boldsymbol{\theta})}{\partial \boldsymbol{\theta}} = \sum_s m(s) \sum_a \frac{\partial \pi(s, a; \boldsymbol{\theta})}{\partial \boldsymbol{\theta}} q_\pi(s, a) \tag{4}$$

*where $m : \mathcal{S} \to [0, \infty)$ is the emphatic weighting, in vector form defined as*

$$\mathbf{m}^\top \stackrel{\text{def}}{=} \mathbf{i}^\top (\mathbf{I} - \mathbf{P}_{\pi,\gamma})^{-1} \tag{5}$$

*where the vector $\mathbf{i} \in \mathbb{R}^{|\mathcal{S}|}$ has entries $\mathbf{i}(s) \stackrel{\text{def}}{=} d_\mu(s) i(s)$ and $\mathbf{P}_{\pi,\gamma} \in \mathbb{R}^{|\mathcal{S}| \times |\mathcal{S}|}$ is the matrix with entries $\mathbf{P}_{\pi,\gamma}(s, s') \stackrel{\text{def}}{=} \sum_a \pi(s, a; \boldsymbol{\theta}) \mathrm{P}(s, a, s') \gamma(s, a, s')$*

*Proof.* First notice that

$$\frac{\partial J_\mu(\boldsymbol{\theta})}{\partial \boldsymbol{\theta}} = \frac{\partial \sum_s \mathbf{i}(s) v_\pi(s)}{\partial \boldsymbol{\theta}} = \sum_s \mathbf{i}(s) \frac{\partial v_\pi(s)}{\partial \boldsymbol{\theta}}$$

Therefore, to compute the gradient of $J_\mu$, we need to compute the gradient of the value function with respect to the policy parameters. A recursive form of the gradient of the value function can be derived, as we show below. Before starting, for simplicity of notation we will use

$$\mathbf{g}(s) = \sum_a \frac{\partial \pi(s, a; \boldsymbol{\theta})}{\partial \boldsymbol{\theta}} q_\pi(s, a)$$

where $\mathbf{g} : \mathcal{S} \to \mathbb{R}^d$. Now let us compute the gradient of the value function.

$$\begin{aligned}
\frac{\partial v_\pi(s)}{\partial \boldsymbol{\theta}} &= \frac{\partial}{\partial \boldsymbol{\theta}} \sum_a \pi(s, a; \boldsymbol{\theta}) q_\pi(s, a) \\
&= \sum_a \frac{\partial \pi(s, a; \boldsymbol{\theta})}{\partial \boldsymbol{\theta}} q_\pi(s, a) + \sum_a \pi(s, a; \boldsymbol{\theta}) \frac{\partial q_\pi(s, a)}{\partial \boldsymbol{\theta}} \\
&= \mathbf{g}(s) + \sum_a \pi(s, a; \boldsymbol{\theta}) \frac{\partial \sum_{s'} \mathrm{P}(s, a, s')(r(s, a, s') + \gamma(s, a, s') v_\pi(s'))}{\partial \boldsymbol{\theta}} \\
&= \mathbf{g}(s) + \sum_a \pi(s, a; \boldsymbol{\theta}) \sum_{s'} \mathrm{P}(s, a, s') \gamma(s, a, s') \frac{\partial v_\pi(s')}{\partial \boldsymbol{\theta}}
\end{aligned} \tag{6}$$

We can simplify this more easily using vector form. Let $\dot{\mathbf{v}}_\pi \in \mathbb{R}^{|\mathcal{S}| \times d}$ be the matrix of gradients (with respect to the policy parameters $\boldsymbol{\theta}$) of $v_\pi$ for each state $s$, and $\mathbf{G} \in \mathbb{R}^{|\mathcal{S}| \times d}$ the matrix where each row corresponding to state $s$ is the vector $\mathbf{g}(s)$. Then

$$\dot{\mathbf{v}}_\pi = \mathbf{G} + \mathbf{P}_{\pi,\gamma} \dot{\mathbf{v}}_\pi \quad \implies \quad \dot{\mathbf{v}}_\pi = (\mathbf{I} - \mathbf{P}_{\pi,\gamma})^{-1} \mathbf{G} \tag{7}$$

Therefore, we obtain

$$\sum_s \mathbf{i}(s) \frac{\partial v_\pi(s)}{\partial \boldsymbol{\theta}} = \mathbf{i}^\top \dot{\mathbf{v}}_\pi \quad = \mathbf{i}^\top (\mathbf{I} - \mathbf{P}_{\pi,\gamma})^{-1} \mathbf{G}$$

$$= \mathbf{m}^\top \mathbf{G}$$

$$= \sum_s m(s) \sum_a \frac{\partial \pi(s,a;\boldsymbol{\theta})}{\partial \boldsymbol{\theta}} q_\pi(s,a)$$

$\square$

We can prove a similar result for the deterministic policy gradient objective, for a deterministic policy, $\pi : \mathcal{S} \to \mathcal{A}$. The objective remains the same, but the space of possible policies is constrained, resulting in a slightly different gradient.

**Theorem 2** (Deterministic Off-policy Policy Gradient Theorem).

$$\frac{\partial J_\mu(\boldsymbol{\theta})}{\partial \boldsymbol{\theta}} = \int_\mathcal{S} m(s) \frac{\partial \pi(s;\boldsymbol{\theta})}{\partial \boldsymbol{\theta}} \left. \frac{\partial q_\pi(s,a)}{\partial a} \right|_{a=\pi(s;\boldsymbol{\theta})} \mathrm{d}s \tag{8}$$

*where $m : \mathcal{S} \to [0,\infty)$ is the emphatic weighting for a deterministic policy, which is the solution to the recursive equation*

$$m(s') \stackrel{\text{def}}{=} d_\mu(s')i(s') + \int_\mathcal{S} \mathrm{P}(s,\pi(s;\boldsymbol{\theta}),s')\gamma(s,\pi(s;\boldsymbol{\theta}),s')m(s)\,\mathrm{d}s \tag{9}$$

The proof is presented in Appendix A.

## 4 Actor-Critic with Emphatic Weightings

In this section, we develop an incremental actor-critic algorithm with emphatic weightings, that uses the above off-policy policy gradient theorem. To perform a gradient ascent update on the policy parameters, the goal is to obtain a sample of the gradient

$$\sum_s m(s) \sum_a \frac{\partial \pi(s,a;\boldsymbol{\theta})}{\partial \boldsymbol{\theta}} q_\pi(s,a). \tag{10}$$

Comparing this expression with the approximate gradient used by OffPAC and subsequent methods (which we refer to as semi-gradient methods) reveals that the only difference is in the weighting of states:

$$\sum_s d_\mu(s) \sum_a \frac{\partial \pi(s,a;\boldsymbol{\theta})}{\partial \boldsymbol{\theta}} q_\pi(s,a). \tag{11}$$

Therefore, we can use standard solutions developed for other actor-critic algorithms to obtain a sample of $\sum_a \frac{\partial \pi(s,a;\boldsymbol{\theta})}{\partial \boldsymbol{\theta}} q_\pi(s,a)$. Explicit details for our off-policy setting are given in Appendix B. The key difficulty is then in estimating $m(s)$ to reweight this gradient, which we address below.

The policy gradient theorem assumes access to the true value function, and provides a Bellman equation that defines the optimal fixed point. However, approximation errors can occur in practice, both in estimating the value function (the critic) and the emphatic weighting. For the critic, we can take advantage of numerous algorithms that improve estimation of value functions, including through the use of $\lambda$-returns to mitigate bias, with $\lambda = 1$ corresponding to using unbiased samples of returns [Sutton, 1988]. For the emphatic weighting, we introduce a similar parameter $\lambda_a \in [0,1]$, that introduces bias but could help reduce variability in the weightings

$$\mathbf{m}_{\lambda_a}^\top = \mathbf{i}^\top (\mathbf{I} - \mathbf{P}_{\pi,\gamma})^{-1}(\mathbf{I} - (1-\lambda_a)\mathbf{P}_{\pi,\gamma}). \tag{12}$$

For $\lambda_a = 1$, we get $\mathbf{m}_{\lambda_a} = \mathbf{m}$ and so get an unbiased emphatic weighting.[4] For $\lambda_a = 0$, the emphatic weighting is simply $\mathbf{i}$, and the gradient with this weighting reduces to the regular off-policy actor critic update [Degris et al., 2012b]. For $\lambda_a = 0$, therefore, we obtain a biased gradient

estimate, but the emphatic weightings themselves are easy to estimate—they are myopic estimates of interest—which could significantly reduce variance when estimating the gradient. Selecting $\lambda_a$ between 0 and 1 could provide a reasonable balance, obtaining a nearly unbiased gradient to enable convergence to a valid stationary point but potentially reducing some variability when estimating the emphatic weighting.

Now we can draw on previous work estimating emphatic weightings incrementally to obtain an emphatically weighted policy gradient. Assume access to an estimate of the gradient $\frac{\partial \pi(s,a;\boldsymbol{\theta})}{\partial \boldsymbol{\theta}} q_\pi(s,a)$, such as the commonly-used estimate: $\rho_t \delta_t \nabla_{\boldsymbol{\theta}} \ln \pi(s,a;\boldsymbol{\theta})$, where $\rho_t$ is the importance sampling ratio (described further in Appendix B), and $\delta_t$ is the temporal difference error, which—as an estimate of the advantage function $a_\pi(s,a) = q_\pi(s,a) - v_\pi(s)$—implicitly includes a state value baseline.

Because this is an off-policy setting, the states $s$ from which we would sample this gradient are weighted according to $d_\mu$. We need to adjust this weighting from $d_\mu(s)$ to $m(s)$. We can do so by using an online algorithm previously derived to obtain a sample $M_t$ of the emphatic weighting

$$ M_t \overset{\text{def}}{=} (1-\lambda_a)i(S_t) + \lambda_a F_t \qquad \qquad \triangleright \; F_t \overset{\text{def}}{=} \gamma_t \rho_{t-1} F_{t-1} + i(S_t) \qquad (13)$$

for $F_0 = 0$. The actor update is then multiplied by $M_t$ to give the emphatically-weighted actor update: $\rho_t M_t \delta_t \nabla_{\boldsymbol{\theta}} \ln \pi(s,a;\boldsymbol{\theta})$. Previous work by Thomas [2014] to remove bias in natural actor-critic algorithms is of interest here, as it suggests weighting actor updates by an accumulating product of discount factors, which can be thought of as an on-policy precursor to emphatic weightings. We prove that our update is an unbiased estimate of the gradient for a fixed policy in Proposition 1. We provide the complete Actor-Critic with Emphatic weightings (ACE) algorithm, with pseudo-code and additional algorithm details, in Appendix B.

**Proposition 1.** *For a fixed policy $\pi$, and with the conditions on the MDP from [Yu, 2015],*

$$ \mathbb{E}_\mu[\rho_t M_t \delta_t \nabla_{\boldsymbol{\theta}} \ln \pi(S_t, A_t; \boldsymbol{\theta})] = \sum_s m(s) \sum_a \frac{\partial \pi(s,a;\boldsymbol{\theta})}{\partial \boldsymbol{\theta}} q_\pi(s,a) $$

*Proof.* Emphatic weightings were previously shown to provide an unbiased estimate for value functions with Emphatic TD. We use the emphatic weighting differently, but can rely on the proof from [Sutton et al., 2016] to ensure that (a) $d_\mu(s)\mathbb{E}_\mu[M_t|S_t = s] = m(s)$ and the fact that (b) $\mathbb{E}_\mu[M_t|S_t = s] = \mathbb{E}_\mu[M_t|S_t = s, A_t, S_{t+1}]$. Using these equalities, we obtain

$\mathbb{E}_\mu[\rho_t M_t \delta_t \nabla_{\boldsymbol{\theta}} \ln \pi(s,a;\boldsymbol{\theta})]$

$\qquad = \sum_s d_\mu(s)\mathbb{E}_\mu[\rho_t M_t \delta_t \nabla_{\boldsymbol{\theta}} \ln \pi(S_t, A_t; \boldsymbol{\theta})|S_t = s]$

$\qquad = \sum_s d_\mu(s)\mathbb{E}_\mu\Big[\mathbb{E}_\mu[\rho_t M_t \delta_t \nabla_{\boldsymbol{\theta}} \ln \pi(S_t, A_t; \boldsymbol{\theta})|S_t = s, A_t, S_{t+1}]\Big] \qquad \triangleright \text{law of total expectation}$

$\qquad = \sum_s d_\mu(s)\mathbb{E}_\mu\Big[\mathbb{E}_\mu[M_t|S_t = s, A_t, S_{t+1}] \; \mathbb{E}_\mu[\rho_t \delta_t \nabla_{\boldsymbol{\theta}} \ln \pi(S_t, A_t; \boldsymbol{\theta})|S_t = s, A_t, S_{t+1}]\Big]$

$\qquad = \sum_s d_\mu(s)\mathbb{E}_\mu[M_t|S_t = s]\mathbb{E}_\mu\Big[\mathbb{E}_\mu[\rho_t \delta_t \nabla_{\boldsymbol{\theta}} \ln \pi(S_t, A_t; \boldsymbol{\theta})|S_t = s, A_t, S_{t+1}]\Big] \qquad \triangleright \text{using (b)}$

$\qquad = \sum_s m(s) \sum_a \frac{\partial \pi(s,a;\boldsymbol{\theta})}{\partial \boldsymbol{\theta}} q_\pi(s,a) \qquad \triangleright \text{using (a).}$

$\hfill \square$

## 5   Experiments

We empirically investigate the utility of using the true off-policy gradient, as opposed to the previous approximation used by OffPAC; the impact of the choice of $\lambda_a$; and the efficacy of estimating emphatic weightings in ACE. We present a toy problem to highlight the fact that OffPAC—which uses an approximate semi-gradient—can converge to suboptimal solutions, even in ideal conditions, whereas ACE—with the true gradient—converges to the optimal solution. We conduct several other experiments on the same toy problem, to elucidate properties of ACE.

## 5.1 The Drawback of Semi-Gradient Updates

We design a world with aliased states to highlight the problem with semi-gradient updates. The toy problem, depicted in Figure 1a, has three states, where S0 is a start state with feature vector $[1, 0]$, and S1 and S2 are aliased, both with feature vector $[0, 1]$. This aliased representation forces the actor to take a similar action in S1 and S2. The behaviour policy takes actions A0 and A1 with probabilities 0.25 and 0.75 in all non-terminal states, so that S0, S1, and S2 will have probabilities 0.5, 0.125, and 0.375 under $d_\mu$. Under this aliasing, the optimal action in S1 and S2 is A0, for the off-policy objective $J_\mu$. The target policy is initialized to take A0 and A1 with probabilities 0.9 and 0.1 in all states, which is near optimal.

We first compared an idealized semi-gradient actor ($\lambda_a = 0$) and gradient actor ($\lambda_a = 1$), with exact value function (critic) estimates. Figures 1b and 1c clearly indicate that the semi-gradient update—which corresponds to an idealized version of the OffPAC update—converges to a suboptimal solution. This occurs even if it is initialized close to the optimal solution, which highlights that the true solution is not even a stationary point for the semi-gradient objective. On the other hand, the gradient solution—corresponding to ACE—increases the objective until converging to optimal. We show below, in Section 5.3 and Figure 5, that this is similarly a problem in the continuous action setting with DPG.

The problem with the semi-gradient updates is made clear from the fact that it corresponds to the $\lambda_a = 0$ solution, which uses the weighting $d_\mu$ instead of $m$. In an expected semi-gradient update, each state tries to increase the probability of the action with the highest action-value. There will be a conflict between the aliased states S1 and S2, since their highest-valued actions differ. If the states are weighted by $\mathbf{d}_\mu$ in the expected update, S1 will appear insignificant to the actor, and the update will increase the probability of A1 in the aliased states. (The ratio between $q_\pi(S1, A0)$ and $q_\pi(S2, A1)$ is not enough to counterbalance this weighting.) However, S1 has an importance that a semi-gradient update overlooks. Taking a suboptimal action at S1 will also reduce $q(S0, A0)$ and, after multiple updates, the actor gradually prefers to take A1 in S0. Eventually, the target policy will be to take A1 at all states, which has a lower objective function than the initial target policy. This experiment suggests why the weight of a state should depend not only on its own share of $\mathbf{d}_\mu$, but also on its predecessors, and the behaviour policy's state distribution is not the proper deciding factor in the competition between S1 and S2.

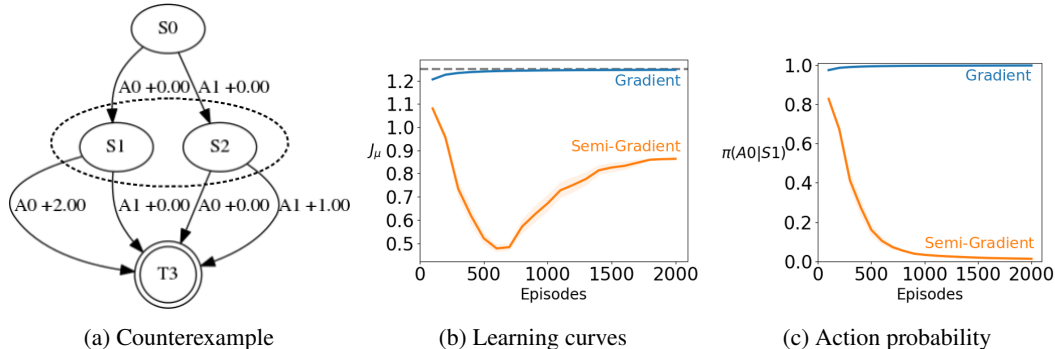

(a) Counterexample          (b) Learning curves          (c) Action probability

Figure 1: (a) A counterexample that identifies suboptimal behaviour when using semi-gradient updates. The semi-gradients converge for the tabular setting [Degris et al., 2012b], but not necessarily under function approximation—such as with the state aliasing in this MDP. S0 is the start state and the terminal state is denoted by T3. S1 and S2 are aliased to the actor. The interest $i(s)$ is set to one for all states. (b) Learning curves comparing semi-gradient updates and gradient updates, averaged over 30 runs with negligible standard error bars. The actor has a softmax output on a linear transformation of features and is trained with a step-size of 0.1 (though results were similar across all the stepsizes tested). The dashed line shows the highest attainable objective function under the aliased representation. (c) The probability of taking A0 at the aliased states, where taking A0 is optimal under the aliased representation.

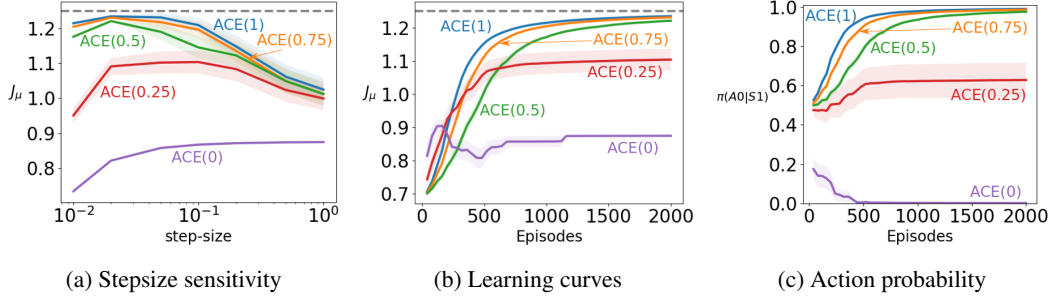

| (a) Stepsize sensitivity | (b) Learning curves | (c) Action probability |

Figure 2: Performance with different values of $\lambda_a$ in the 3-state MDP, averaged over 30 runs. (a) ACE performs well, for a range of stepsizes and even $\lambda_a$ that gets quite small. (b) For $\lambda_a = 0$, which corresponds to OffPAC, the algorithm decreases performance, to get to the suboptimal fixed point. Even with as low a value as $\lambda_a = 0.25$, ACE improves the value from the starting point, but does converge to a worse solution than $\lambda_a \geq 0.5$. The learning curves correspond to each algorithm's best step-size. (c) The optimal behaviour is to take A0 with probability 1, in the aliased states. ACE(0)—corresponding to OffPAC—quickly converges to the suboptimal solution of preferring the optimal action for S2 instead of S1. Even with $\lambda_a$ just a bit higher than 0, convergence is to a more reasonable solution, preferring the optimal action the majority of the time.

## 5.2 The Impact of the Trade-Off Parameter

The parameter $\lambda_a$ in (12) has the potential to trade off bias and variance. For $\lambda_a = 0$, the bias can be significant, as shown in the previous section. A natural question is how the bias changes as $\lambda_a$ decreases from 1 to 0. There is unlikely to be significant variance reduction—it is a low variance domain—but we can nonetheless gain some insight into bias.

We repeated the experiment in 5.1, with $\lambda_a$ chosen from $\{0, 0.25, 0.5, 0.75, 1\}$ and the step-size chosen from $\{0.01, 0.02, 0.05, 0.1, 0.2, 0.5, 1\}$. To highlight the rate of learning, the actor parameters are now initialized to zero. Figure 2 summarizes the results. As shown in Figure 2a, with $\lambda_a$ close to one, a small and carefully-tuned step-size is needed to make the most of the method. As $\lambda_a$ decreases, the actor is able to learn with higher step-sizes and increases the objective function faster. The quality of the final solution, however, deteriorates with small values of $\lambda_a$ since the actor follows biased gradients. Even for surprisingly small $\lambda_a = 0.5$ the actor converged to optimal, and even $\lambda_a = 0.25$ produced a much more reasonable solution than $\lambda_a = 0$.

We ran a similar experiment, this time using value estimates from a critic trained with Gradient TD, called GTD($\lambda$) [Maei, 2011] to examine whether the impact of $\lambda_a$ values with the actual (non-idealized) ACE algorithm persists in an actor-critic architecture. The step-size $\alpha_v$ was chosen from $\{10^{-5}, 10^{-4}, 10^{-3}, 10^{-2}, 10^{-1}, 10^0\}$, $\alpha_w$ was chosen from $\{10^{-10}, 10^{-8}, 10^{-6}, 10^{-4}, 10^{-2}\}$, and $\{0, 0.5, 1.0\}$ was the set of candidate values of $\lambda$ for the critic. The results in Figure 3 show that, as before, even relatively low $\lambda_a$ values can still get close to the optimal solution. However, semi-gradient updates, corresponding to $\lambda_a = 0$, still find a suboptimal policy.

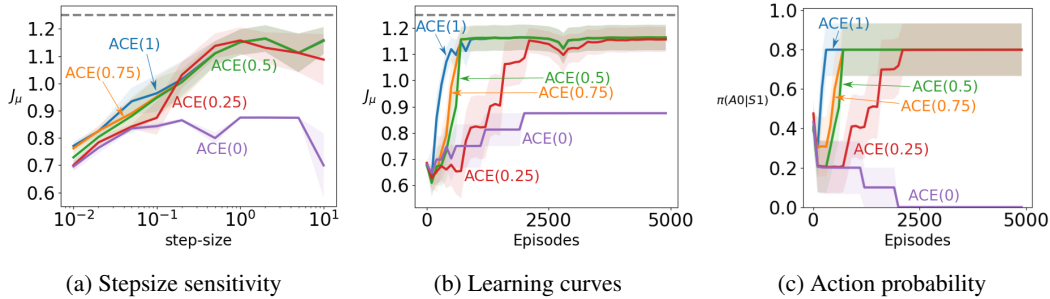

| (a) Stepsize sensitivity | (b) Learning curves | (c) Action probability |

Figure 3: Performance of ACE with a GTD($\lambda$) critic and different values of $\lambda_a$ in the 3-state MDP. The results are averaged over 10 runs. The outcomes are similar to Figure 2, though noisier due to learning the critic rather than using the true critic.

## 5.3 Challenges in Estimating the Emphatic Weightings

We have been using an online algorithm to estimate the emphatic weightings. There can be different sources of inaccuracy in these approximations. First, the estimate depends on importance sampling ratios of previous actions in the trajectory. This can result in high variance if the behaviour policy and the target policy are not close. Secondly, derivation of the online algorithm assumes a fixed target policy [Sutton et al., 2016], while the actor is updated at every time step. Therefore, the approximation is less reliable in long trajectories, as it is partly decided by older target policies in the beginning of the trajectory. We designed experiments to study the efficacy of these estimates in an aliased task with more states.

The first environment, shown in Figure 6 in the appendix, is an extended version of the previous MDP with two long chains before the aliased states. The addition of new states makes the trajectories considerably longer, allowing errors to build up in emphatic weighting estimates. The actor is initialized with zero weights and uses true state values in its updates. The behaviour policy takes A0 with probability 0.25 and A1 with probability 0.75 in all non-terminal states. The actor's step-size is picked from $\{5 \cdot 10^{-5}, 10^{-4}, 2 \cdot 10^{-4}, 5 \cdot 10^{-4}, 10^{-3}, 2 \cdot 10^{-3}, 5 \cdot 10^{-3}, 10^{-2}\}$. We also trained an actor, called True-ACE, that uses true emphatic weightings for the current target policy and behaviour policy, computed at each timestep. The performance of True-ACE is included here for the sake of comparison, and computing the exact emphatic weightings is not generally possible in an unknown environment. The results in Figure 4 show that, even though performance improves as $\lambda_a$ is increased, there is a gap between ACE with $\lambda_a = 1$ and True-ACE. This shows the inaccuracies pointed out above indeed disturb the updates in long trajectories.

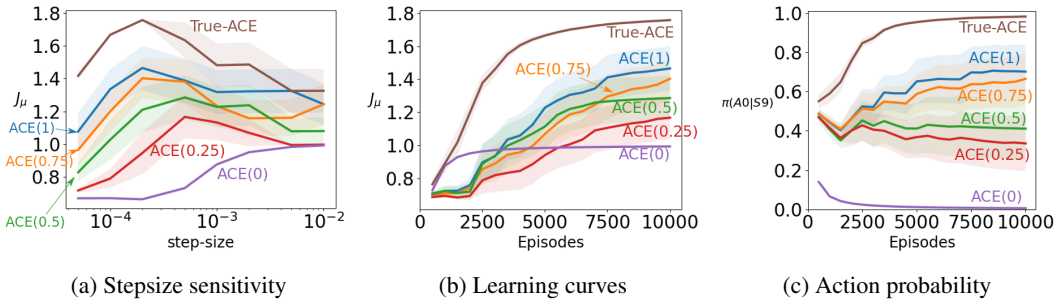

(a) Stepsize sensitivity      (b) Learning curves      (c) Action probability

Figure 4: Performance of ACE with different values of $\lambda_a$ and True-ACE on the 11-state MDP. The results are averaged over 10 runs. Unlike Figure 2, the methods now have more difficulty getting near the optimal solution, though ACE with larger $\lambda_a$ does still clearly get a significantly better solution that $\lambda_a = 0$.

The second environment is similar to Figure 1a, but with one continuous unbounded action. Taking action with value $a$ at S0 will result in a transition to S1 with probability $1 - \sigma(a)$ and a transition to S2 with probability $\sigma(a)$, where $\sigma$ denotes the logistic sigmoid function. For all actions from S0, the reward is zero. From S1 and S2, the agent can only transition to the terminal state, with reward $2\sigma(-a)$ and $\sigma(a)$ respectively. The behaviour policy takes actions drawn from a Gaussian distribution with mean 1.0 and variance 1.0.

Because the environment has continuous actions, we can include both stochastic and deterministic policies, and so can include DPG in the comparison. DPG is built on the semi-gradient, like OffPAC. We compare to DPG with Emphatic weightings (DPGE), with the true emphatic weightings rather than estimated ones. We compare to True-DPGE to avoiding confounding factors of estimating the emphatic weighting, and focus the investigation on if DPG converges to a suboptimal solution. Estimation of the emphatic weightings for a deterministic target policy is left for future work. The stochastic actor in ACE has a linear output unit and a softplus output unit to represent the mean and the standard deviation of a Gaussian distribution. All actors are initialized with zero weights.

Figure 5 summarizes the results. The first observation is that DPG demonstrates suboptimal behaviour similar to OffPAC. As training goes on, DPG prefers to take positive actions in all states, because S2 is updated more often. This problem goes away in True-DPGE. The emphatic weightings emphasize updates in S1 and, thus, the actor gradually prefers negative actions and surpasses DPG in performance. Similarly, True-ACE learns to take negative actions but, being a stochastic policy, it

cannot achieve True-DPGE's performance on this domain. ACE with different $\lambda_a$ values, however, cannot outperform DPG, and this result suggests that an alternative to importance sampling ratios is needed to extend ACE to continuous actions.

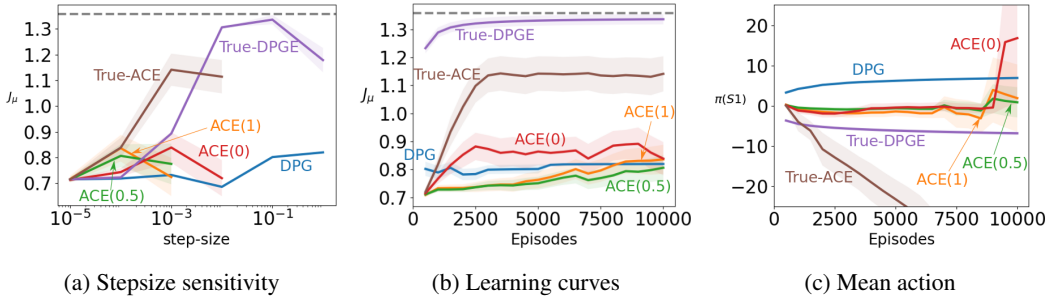

(a) Stepsize sensitivity        (b) Learning curves        (c) Mean action

Figure 5: Performance of ACE with different values of $\lambda_a$, True-ACE, DPG, and True-DPGE on the continuous action MDP. The results are averaged over 30 runs. For continuous actions, the methods have even more difficulty getting to the optimal solutions, given by True-DPGE and True-ACE, though the action selection graphs suggest that ACE for higher $\lambda_a$ is staying nearer the optimal action selection than ACE(0) and DPG.

## 6   Conclusions and Future Work

In this paper we proved the off-policy policy gradient theorem, using emphatic weightings. The result is generally applicable to any differentiable policy parameterization. Using this theorem, we derived an off-policy actor-critic algorithm that follows the gradient of the objective function, as opposed to previous method like OffPAC and DPG that followed an approximate semi-gradient. We designed a simple MDP to highlight issues with existing methods—namely OffPAC and DPG— particularly highlighting that the stationary points of these semi-gradient methods for this problem do not include the optimal solution. Our algorithm, called Actor-Critic with Emphatic Weightings, on the other hand, which follows the gradient, reaches the optimal solution, both for an idealized setting given the true critic and when learning the critic. We conclude with a result suggesting that more work needs to be done to effectively estimate emphatic weightings, and that important next steps for developing Actor-Critic algorithm for the off-policy setting are to improve estimation of these weightings.

## 7   Acknowledgements

The authors would like to thank Alberta Innovates for funding the Alberta Machine Intelligence Institute and by extension this research. We would also like to thank Hamid Maei, Susan Murphy, and Rich Sutton for their helpful discussions and insightful comments.

## Footnotes

[2]See B. Errata in Degris et al. [2012b] for the clarification that the theorem only applies to tabular policy representations.

[3]Note that the statement in the paper is stronger, but in an errata published by the authors, they highlight an error in the proof. Consequently, the result is only correct for the tabular setting.

[4]Note that the original emphatic weightings [Sutton et al., 2016] use $\lambda = 1 - \lambda_a$. This is because their emphatic weightings are designed to balance bias introduced from using $\lambda$ for estimating value functions: larger $\lambda$ means the emphatic weighting plays less of a role. For this setting, we want larger $\lambda_a$ to correspond to the full emphatic weighting (the unbiased emphatic weighting), and smaller $\lambda_a$ to correspond to a more biased estimate, to better match the typical meaning of such trace parameters.

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
