[Supplementary Material · ace_full.pdf]

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

# A  Proof of Deterministic Off-Policy Policy Gradient Theorem

## A.1  Assumptions

We make the following assumptions on the MDP:

**Assumption 1.** $P(s, a, s'), r(s, a, s'), \gamma(s, a, s'), \pi(s; \boldsymbol{\theta})$ *and their derivatives are continuous in all variables* $s, a, s', \boldsymbol{\theta}$.

**Assumption 2.** $\mathcal{S}$ *is a compact set of* $\mathbb{R}^d$.

**Assumption 3.** *The policy* $\pi$ *and discount* $\gamma$ *are such that for* $\mathbf{P}_{\pi,\gamma}(s, s') = \int_{\mathcal{A}} \pi(s, a) \gamma(s, a, s') P(s, a, s') da$, *the inverse kernel of* $\delta(s, s') - \mathbf{P}_{\pi,\gamma}(s, s')$ *exists.*

Under Assumption 1, $v_{\pi_{\boldsymbol{\theta}}}(s)$ and $\frac{\partial v_{\pi_{\boldsymbol{\theta}}}(s)}{\partial \boldsymbol{\theta}}$ are continuous functions of $\boldsymbol{\theta}$ and $s$. Together, Assumptions 1 and 2 imply that $\left\| \frac{\partial v_{\pi_{\boldsymbol{\theta}}}(s)}{\partial \boldsymbol{\theta}} \right\|, \left\| \frac{\partial \pi(s; \boldsymbol{\theta})}{\partial \boldsymbol{\theta}} \right\|$, and $\left\| \frac{\partial q_{\pi_{\boldsymbol{\theta}}}(s, a)}{\partial a} \Big|_{a=\pi(s; \boldsymbol{\theta})} \right\|$ are bounded functions of $s$, which allows us to switch the order of integration and differentiation, and the order of multiple integrations later in the proof.

## A.2  Proof of Theorem 2

*Proof.* We start by deriving a recursive form for the gradient of the value function with respect to the policy parameters:

$$
\frac{\partial v_\pi(s)}{\partial \boldsymbol{\theta}} = \frac{\partial}{\partial \boldsymbol{\theta}} q_\pi(s, \pi(s; \boldsymbol{\theta}))
$$

$$
= \frac{\partial}{\partial \boldsymbol{\theta}} \int_{\mathcal{S}} P(s, \pi(s; \boldsymbol{\theta}), s') \Big( r(s, \pi(s; \boldsymbol{\theta}), s') + \gamma(s, \pi(s; \boldsymbol{\theta}), s') v_\pi(s') \Big) \, ds'
$$

$$
= \int_{\mathcal{S}} \frac{\partial}{\partial \boldsymbol{\theta}} \Big( P(s, \pi(s; \boldsymbol{\theta}), s') \big( r(s, \pi(s; \boldsymbol{\theta}), s') + \gamma(s, \pi(s; \boldsymbol{\theta}), s') v_\pi(s') \big) \Big) \, ds' \quad (14)
$$

where in (14) we used the Leibniz integral rule to switch the order of integration and differentiation. We proceed with the derivation using the product rule:

$$
= \int_{\mathcal{S}} \frac{\partial}{\partial \boldsymbol{\theta}} P(s, \pi(s; \boldsymbol{\theta}), s') \Big( r(s, \pi(s; \boldsymbol{\theta}), s') + \gamma(s, \pi(s; \boldsymbol{\theta}), s') v_\pi(s') \Big)
$$

$$
+ P(s, \pi(s; \boldsymbol{\theta}), s') \frac{\partial}{\partial \boldsymbol{\theta}} \Big( r(s, \pi(s; \boldsymbol{\theta}), s') + \gamma(s, \pi(s; \boldsymbol{\theta}), s') v_\pi(s') \Big) \, ds'
$$

$$
= \int_{\mathcal{S}} \frac{\partial \pi(s; \boldsymbol{\theta})}{\partial \boldsymbol{\theta}} \frac{\partial P(s, a, s')}{\partial a} \Big|_{a=\pi(s; \boldsymbol{\theta})} \Big( r(s, \pi(s; \boldsymbol{\theta}), s') + \gamma(s, \pi(s; \boldsymbol{\theta}), s') v_\pi(s') \Big)
$$

$$
+ P(s, \pi(s; \boldsymbol{\theta}), s') \left( \frac{\partial}{\partial \boldsymbol{\theta}} r(s, \pi(s; \boldsymbol{\theta}), s') + \frac{\partial}{\partial \boldsymbol{\theta}} \gamma(s, \pi(s; \boldsymbol{\theta}), s') v_\pi(s') + \gamma(s, \pi(s; \boldsymbol{\theta}), s') \frac{\partial}{\partial \boldsymbol{\theta}} v_\pi(s') \right) \, ds'
$$

$$
= \int_{\mathcal{S}} \frac{\partial \pi(s; \boldsymbol{\theta})}{\partial \boldsymbol{\theta}} \frac{\partial P(s, a, s')}{\partial a} \Big|_{a=\pi(s; \boldsymbol{\theta})} \Big( r(s, \pi(s; \boldsymbol{\theta}), s') + \gamma(s, \pi(s; \boldsymbol{\theta}), s') v_\pi(s') \Big)
$$

$$
+ P(s, \pi(s; \boldsymbol{\theta}), s') \left( \frac{\partial \pi(s; \boldsymbol{\theta})}{\partial \boldsymbol{\theta}} \frac{\partial r(s, a, s')}{\partial a} \Big|_{a=\pi(s; \boldsymbol{\theta})} + \frac{\partial \pi(s; \boldsymbol{\theta})}{\partial \boldsymbol{\theta}} \frac{\partial \gamma(s, a, s')}{\partial a} \Big|_{a=\pi(s; \boldsymbol{\theta})} v_\pi(s') \right) \, ds'
$$

$$
+ \int_{\mathcal{S}} P(s, \pi(s; \boldsymbol{\theta}), s') \gamma(s, \pi(s; \boldsymbol{\theta}), s') \frac{\partial}{\partial \boldsymbol{\theta}} v_\pi(s') \, ds'
$$

$$
= \int_{\mathcal{S}} \frac{\partial \pi(s; \boldsymbol{\theta})}{\partial \boldsymbol{\theta}} \left( \frac{\partial P(s, a, s')}{\partial a} \Big|_{a=\pi(s; \boldsymbol{\theta})} \Big( r(s, \pi(s; \boldsymbol{\theta}), s') + \gamma(s, \pi(s; \boldsymbol{\theta}), s') v_\pi(s') \Big) \right.
$$

$$
\left. + P(s, \pi(s; \boldsymbol{\theta}), s') \frac{\partial}{\partial a} \Big( r(s, a, s') + \gamma(s, a, s') v_\pi(s') \Big) \Big|_{a=\pi(s; \boldsymbol{\theta})} \right) \, ds'
$$

$$
+ \int_{\mathcal{S}} P(s, \pi(s; \boldsymbol{\theta}), s') \gamma(s, \pi(s; \boldsymbol{\theta}), s') \frac{\partial}{\partial \boldsymbol{\theta}} v_\pi(s') \, ds'
$$

$$= \frac{\partial \pi(s; \boldsymbol{\theta})}{\partial \boldsymbol{\theta}} \int_{\mathbb{S}} \frac{\partial}{\partial a} \left( P(s, a, s') \left( r(s, a, s') + \gamma(s, a, s') v_\pi(s') \right) \right) \Big|_{a=\pi(s;\boldsymbol{\theta})} \mathrm{d}s'$$

$$+ \int_{\mathbb{S}} P(s, \pi(s; \boldsymbol{\theta}), s') \gamma(s, \pi(s; \boldsymbol{\theta}), s') \frac{\partial v_\pi(s')}{\partial \boldsymbol{\theta}} \mathrm{d}s'$$

$$= \frac{\partial \pi(s; \boldsymbol{\theta})}{\partial \boldsymbol{\theta}} \frac{\partial q_\pi(s, a)}{\partial a} \Big|_{a=\pi(s;\boldsymbol{\theta})} + \int_{\mathbb{S}} P(s, \pi(s; \boldsymbol{\theta}), s') \gamma(s, \pi(s; \boldsymbol{\theta}), s') \frac{\partial v_\pi(s')}{\partial \boldsymbol{\theta}} \mathrm{d}s' \tag{15}$$

For simplicity of notation, we will write (15) as:

$$\frac{\partial v_\pi(s)}{\partial \boldsymbol{\theta}} = g(s) + \int_{\mathbb{S}} \mathbf{P}_{\pi,\gamma}(s, s') \frac{\partial v_\pi(s')}{\partial \boldsymbol{\theta}} \mathrm{d}s' \tag{16}$$

since $P(s, \pi(s; \boldsymbol{\theta}), s') \gamma(s, \pi(s; \boldsymbol{\theta}), s')$ is a function of $s$ and $s'$ for a fixed deterministic policy.

Note that we can write $\frac{\partial v_\pi(s)}{\partial \boldsymbol{\theta}}$ as an integral transform using the delta function:

$$\frac{\partial v_\pi(s)}{\partial \boldsymbol{\theta}} = \int_{\mathbb{S}} \delta(s, s') \frac{\partial v_\pi(s')}{\partial \boldsymbol{\theta}} \mathrm{d}s' \tag{17}$$

Plugging (17) into the left-hand side of (16), we obtain:

$$\int_{\mathbb{S}} \delta(s, s') \frac{\partial v_\pi(s')}{\partial \boldsymbol{\theta}} \mathrm{d}s' = g(s) + \int_{\mathbb{S}} \mathbf{P}_{\pi,\gamma}(s, s') \frac{\partial v_\pi(s')}{\partial \boldsymbol{\theta}} \mathrm{d}s'$$

$$\implies \int_{\mathbb{S}} \left( \delta(s, s') - \mathbf{P}_{\pi,\gamma}(s, s') \right) \frac{\partial v_\pi(s')}{\partial \boldsymbol{\theta}} \mathrm{d}s' = g(s)$$

$$\implies \frac{\partial v_\pi(s)}{\partial \boldsymbol{\theta}} = \int_{\mathbb{S}} k(s, s') g(s') \mathrm{d}s' \tag{18}$$

where $k(s, s')$ is the inverse kernel of $\delta(s, s') - \mathbf{P}_{\pi,\gamma}(s, s')$. Now, using a continuous version of the off-policy objective defined in (3), we have:

$$\frac{\partial J_\mu(\boldsymbol{\theta})}{\partial \boldsymbol{\theta}} = \frac{\partial}{\partial \boldsymbol{\theta}} \int_{\mathbb{S}} d_\mu(s) i(s) v_\pi(s) \mathrm{d}s$$

$$= \int_{\mathbb{S}} \frac{\partial v_\pi(s)}{\partial \boldsymbol{\theta}} d_\mu(s) i(s) \mathrm{d}s$$

$$= \int_{\mathbb{S}} \int_{\mathbb{S}} k(s, s') g(s') \mathrm{d}s' \, d_\mu(s) i(s) \mathrm{d}s$$

$$= \int_{\mathbb{S}} \int_{\mathbb{S}} k(s, s') d_\mu(s) i(s) \mathrm{d}s \, g(s') \mathrm{d}s' \tag{19}$$

where in (19) we used Fubini's theorem to switch the order of integration.

Now, we convert the recursive definition of emphatic weightings for deterministic policies over continuous state-action spaces into a non-recursive form. Recall the definition:

$$m(s') = d_\mu(s') i(s') + \int_{\mathbb{S}} \mathbf{P}_{\pi,\gamma}(s, s') m(s) \mathrm{d}s \tag{20}$$

Again, we can write $m(s')$ as an integral transform using the delta function:

$$m(s') = \int_{\mathbb{S}} \delta(s, s') m(s) \mathrm{d}s \tag{21}$$

Plugging (21) into the left-hand side of (20), we obtain:

$$\int_{\mathbb{S}} \delta(s, s') m(s) \mathrm{d}s = d_\mu(s') i(s') + \int_{\mathbb{S}} \mathbf{P}_{\pi,\gamma}(s, s') m(s) \mathrm{d}s$$

$$\implies \int_{\mathbb{S}} (\delta(s, s') - \mathbf{P}_{\pi,\gamma}(s, s')) m(s) \mathrm{d}s = d_\mu(s') i(s')$$

$$\implies m(s') = \int_{\mathbb{S}} k(s, s') d_\mu(s) i(s) \mathrm{d}s \tag{22}$$

where $k(s, s')$ is again the inverse kernel of $\delta(s, s') - \mathbf{P}_{\pi,\gamma}(s, s')$. Plugging equation (22) into (19) yields:

$$
\begin{aligned}
\frac{\partial J_\mu(\boldsymbol{\theta})}{\partial \boldsymbol{\theta}} &= \int_{\mathbb{S}} m(s')g(s')\,\mathrm{d}s' \\
&= \int_{\mathbb{S}} m(s)\frac{\partial \pi(s;\boldsymbol{\theta})}{\partial \boldsymbol{\theta}}\,\frac{\partial q_\pi(s, a)}{\partial a}\bigg|_{a=\pi(s;\boldsymbol{\theta})}\,\mathrm{d}s
\end{aligned}
$$

$\square$

## B  Algorithm details

Observed states and actions are sampled according to the behaviour policy $\mu$. Notice the inner sum over actions in equation 10 is not weighted by any distribution, and will therefore be skewed by sampling the actions according to $\mu$. One option for solving this problem is to explicitly sum the gradient over actions. This has the added benefit of reducing variability, but for many actions could be impractical. An alternative is to modify the sampling distribution using importance sampling:

$$
\begin{aligned}
\sum_a \frac{\partial \pi(s, a;\boldsymbol{\theta})}{\partial \boldsymbol{\theta}}q_\pi(s, a) &= \sum_a \mu(s, a)\frac{1}{\mu(s, a)}\frac{\partial \pi(s, a;\boldsymbol{\theta})}{\partial \boldsymbol{\theta}}q_\pi(s, a) \\
&= \sum_a \mu(s, a)\frac{\pi(s, a;\boldsymbol{\theta})}{\mu(s, a)}\frac{1}{\pi(s, a;\boldsymbol{\theta})}\frac{\partial \pi(s, a;\boldsymbol{\theta})}{\partial \boldsymbol{\theta}}q_\pi(s, a) \\
&= \sum_a \mu(s, a)\rho(s, a;\boldsymbol{\theta})\frac{\partial \ln \pi(s, a;\boldsymbol{\theta})}{\partial \boldsymbol{\theta}}q_\pi(s, a)
\end{aligned}
$$

where the last step follows from the fact that the derivative of $\ln y$ is $\frac{1}{y}$. Now this reweighted gradient can be sampled by sampling states according to $\mathbf{d}_\mu$, actions according to $\mu$ and then weighting the update $\rho(s, a;\boldsymbol{\theta})\frac{\partial \ln \pi(s, a;\boldsymbol{\theta})}{\partial \boldsymbol{\theta}}q_\pi(s, a)$ with $M_t$ as in equation 13. The resulting update on each step, without any additional variance reduction, for a particular action is

$$
\boldsymbol{\theta} \leftarrow \boldsymbol{\theta} + \alpha\rho_t M_t \frac{\partial \ln \pi(s, a;\boldsymbol{\theta})}{\partial \boldsymbol{\theta}}q_\pi(s, a)
$$

and when summed over all actions, or a subset of actions, is

$$
\boldsymbol{\theta} \leftarrow \boldsymbol{\theta} + \alpha M_t \sum_{b\in\mathcal{A}} \pi(s, b;\boldsymbol{\theta})\frac{\partial \ln \pi(s, b;\boldsymbol{\theta})}{\partial \boldsymbol{\theta}}q_\pi(s, b)
$$

The second approach is generally more suitable, as it avoids potentially high-variance importance sampling ratios. The first approach, though, is necessary when only a value function is estimated, as described in the body of the paper. In the next section we consider other approaches to reduce variance of this update. The final ACE algorithm incorporating these variance reduction techniques is summarized in Algorithm 1.

### B.1  Incorporating baselines

To reduce the variance of the sampled gradient, it is common to include a baseline as a form of control variate. The baseline $b : \mathbb{S} \to \mathbb{R}$ is incorporated into the update as

$$
\sum_a \frac{\partial \pi(s, a;\boldsymbol{\theta})}{\partial \boldsymbol{\theta}}[q_\pi(s, a) - b(s)] \tag{23}
$$

The typical choice of baseline is the value function $b(s) = v_\pi(s)$, because

$$
\begin{aligned}
\sum_a \frac{\partial \pi(s, a;\boldsymbol{\theta})}{\partial \boldsymbol{\theta}}v_\pi(s) &= v_\pi(s)\frac{\partial}{\partial \boldsymbol{\theta}}\sum_a \pi(s, a;\boldsymbol{\theta}) \\
&= v_\pi(s)\frac{\partial}{\partial \boldsymbol{\theta}}1 \\
&= 0
\end{aligned}
$$

---

**Algorithm 1** Emphatic Actor Critic

---

Initialize weights for actor $\boldsymbol{\theta}$ to zero
Initialize emphatic weightings for actor: $F_{-1} = 0$
Initialize importance sampling ratios: $\rho_{-1} = 1$
Suggested (default) settings of parameters: $i_t = 1$, $\lambda_a = 0.9$
Obtain initial feature vector $\mathbf{x}_0$
**repeat**
    Choose an action $a_t$ according to $\mu(\mathbf{x}_t, \cdot)$
    Observe reward $r_{t+1}$, next state vector $\mathbf{x}_{t+1}$ and $\gamma_{t+1}$
    Update critic $\hat{V}_t$ (and potentially $\hat{Q}_t$) with any value function learning algorithm
    $\rho_t \leftarrow \frac{\pi(\mathbf{x}_t, a_t)}{\mu(\mathbf{x}_t, a)}$
    $F_t \leftarrow \rho_{t-1} \gamma_t F_{t-1} + i_t$
    $M_t \leftarrow (1 - \lambda_{a,t}) i_t + \lambda_{a,t} F_t$
    **if** Only learned $\hat{V}_t$ **then**
        $\boldsymbol{\psi}_t \leftarrow \nabla_{\boldsymbol{\theta}} \ln \pi(\mathbf{x}_t, a_t; \boldsymbol{\theta})$
        $\delta_t \leftarrow r_{t+1} + \gamma_{t+1} \hat{V}_t(\mathbf{x}_{t+1}) - \hat{V}_t(\mathbf{x}_t)$
        $\boldsymbol{\theta} \leftarrow \boldsymbol{\theta} + \alpha_t \rho_t M_t \delta_t \boldsymbol{\psi}_t$
    **else**
        **for** $b \in \mathcal{A}$ or a randomly sampled subset **do**
            $\boldsymbol{\psi}_t \leftarrow \nabla_{\boldsymbol{\theta}} \ln \pi(s, b; \boldsymbol{\theta})$
            $\boldsymbol{\theta} \leftarrow \boldsymbol{\theta} + \alpha_t M_t \pi(s, b; \boldsymbol{\theta})(\hat{Q}_t(\mathbf{x}_t, b) - \hat{V}_t(\mathbf{x}_t)) \boldsymbol{\psi}_t$
**until** agent done interaction with environment

---

However, the variance of $q_\pi(s, a) - v_\pi(s)$ is lower because $v_\pi(s)$ is correlated with $q_\pi(s, a)$ [Williams, 1992].

Estimates of 23 can then be computed in at least three ways. The first is to simply estimate $v_\pi$ and $q_\pi$. The second is to estimate $v_\pi$, and then estimate the advantage function $a_\pi(s, a) = q_\pi(s, a) - v_\pi(s)$. The advantage function can be updated using $\delta_t$, which compares the value for the given action $a$ from state $s$, $r(s, a) + \gamma(s, a, S_{t+1}) v_\pi(S_{t+1})$, to the value at state s, $v_\pi(s)$. The third is to again use $\delta_t$, but to avoid computing $q_\pi$ altogether. For this third approach, $r(s, a) + \gamma(s, a, S_{t+1}) v_\pi(S_{t+1})$ is used as an approximation of $q_\pi(s, a)$. To improve this approximation, a $\lambda$-return could be used, which then causes traces of gradients of policy parameters to be used (see Algorithm 1 [Degris et al., 2012b]). Because the traces for the policy parameters are different for ACE, we do not include this additional trace. It could be added if only $v_\pi$ is learned, but we anticipate that in addition to the emphatic weighting, this would induce too much variability in the algorithm. When using only $v_\pi$, therefore, we set this trace parameter to zero and $q_\pi(s_t, a_t) - v_\pi(s_t)$ is approximated with $\delta_t$.

Figure 6: An 11-state MDP that makes estimating the emphatic weightings difficult. S0 is the start state and the terminal state is denoted by T11. S9 and S10 are aliased to the actor.