[Reviews · NeurIPS 2018]

Reviewer 1



The paper formulates and proves a policy gradient theorem in the off-policy setting. The derivation is based on emphatic weighting of the states. Based on the introduced theorem, an actor-critic algorithm, termed ACE, is further proposed. The algorithm requires computing policy gradient updates that depend on the emphatic weights. Computing low-variance estimates of the weights is non-trivial, and the authors introduce a relaxed version of the weights that interpolate between the off-policy actor-critic (Degris et al., 2012) and the unbiased (but high variance) estimator; the introduced estimator can be computed incrementally. The paper is well written and makes an interesting contribution. The authors might further improve the paper by emphasizing about what are the specific novel contributions of their paper (both on theory and algorithmic side), as it is unclear in some parts whether a certain proposal is novel or has been adapted from related/previous work and combined with other concepts in a novel way. (E.g., Thm. 1 seems straightforward so long as the notion of emphatic weights is introduced; would be helpful if the authors can clarify whether/why it hasn't been derived before or whether it was studied in a different context.) Additional comments and questions: - Even thought it is stated at the beginning of section 2, I would recommend making it clear in the statements of the theorems that the state and action spaces are assumed to be finite. Eq. 7 implicitly makes use of that. In case of continuous state space, one would have to solve the Fredholm integral equation of second kind instead, which I believe is not as straightforward. - Eq. 11 proposes a very specific adjustment to the spectrum of the (I - P_{\pi, \gamma}). The authors analyze empirically the learning behavior of ACE with different values of lambda_a. I am curious whether the spectrum of (I - P_{\pi, \gamma}), which should be based on the properties of the MDP and the policy, has any effect on the choice of the hyperparameter? - (On-policy) policy gradients are widely used beyond discrete and finite-state MDPs. It is unclear whether ACE is as widely applicable. If not what are the key challenges that need to be addressed? I would like the authors to comment on that.

Reviewer 2



This paper derives the exact gradient for off-policy policy gradient with emphatic weights, and use an iterative algorithm to provide an unbiased estimator of the weight. The new algorithm corrects the gradient used in the previous OffPAC algorithm and showed significant improvement in the experiments. I think the paper is interesting and well written, and the algorithm is novel based on my knowledge. Some comments: 1. I have questions about the objective function (3). Why use this objective for off-policy policy gradient? Based on the equation, this objective seems to take behavior policy for time t->\infty, and then take model policy for the rest. I cannot get the intuition why this objective is useful and how it related to the online policy gradient objective. 2. How to prove that the limiting distribution d_\mu(s) exists? 3. What is \rho and \delta_t in Line 131? 4. The intuition seems that larger \lambda_a increases the variance of the iterative estimate of emphatic weights. It would be great to derive the relation between the variance and lambda_a. It would also be good to show that in the experiments. 5. In proposition 1, what is the expectation over in the LHS of equation.

Reviewer 3



After rebuttal: The author addresses some of my questions in the rebuttal. Overall, I think the paper can use better presentation of the theorems and contrast with previous results on semi-gradient theorems. For now the theory might be quite limited in its scalability to larger problems, it might be interesting to explore further how one can combine it with scalable methods in the future. I increase my rating a bit. ====== Summary: The paper proposes a new off-policy gradient form as well as a new procedure to compute the gradient using emphatic weighting. The paper also empirically compares the proposed method with DPG algorithms on a simple three state MDP problem, showing that the proposed method can learn optimal policies while DPG is stuck at local optimal. Questions: 1. Line 96. dot v_s is the gradient of v_s with respect to states or parameters? The states are discrete here so I presume it is parameters? Can authors write down the mathematical form of the gradient in the paper. 2. The author should make it more clear that the LHS of Proposition 1 corresponds to the gradient of interest. 3. What is semi-gradient? The paper did not seem to clarify what this definition is. For example, the update of DPG does follow the gradient (with some technical details to drop an additional term), does the ‘semi-‘ come from dropping such a term? General Advice: 1. Previous work. Though the paper mentions previous works on offPAC and DPG on off-policy optimization, the comparison is not made obvious and clear. Importantly, the paper did not write the mathematical equations for DPG (for example), and make it a bit difficult for direct comparison. 2. Notations and presentation. There is much notation in the paper that is not quite directly clear. I think it definitely helps to clarify interpretations of some terms in (for example) LHS of proposition1. 3. Experiments are a bit weak. The experiments could include probably some more conventional MDP systems to illustrate that the new method does not deteriorate compared with DPG/offPAC because of the difficulty in weight estimate. Current experiments only consider the three MDP system and seem a bit weak. Also, is it possible to carry out function approximation in such a setup to make the algorithm scale and applicable to more realistic system?